# Sexual Harassment by Patients, Clients, and Residents: Investigating Its Prevalence, Frequency and Associations with Impaired Well-Being among Social and Healthcare Workers in Germany

**DOI:** 10.3390/ijerph18105198

**Published:** 2021-05-13

**Authors:** Mareike Adler, Sylvie Vincent-Höper, Claudia Vaupel, Sabine Gregersen, Anja Schablon, Albert Nienhaus

**Affiliations:** 1Department of Occupational Medicine, Hazardous Substances and Public Health, German Social Accident Insurance Institution for Health and Welfare Services, 22089 Hamburg, Germany; Claudia.Vaupel@bgw-online.de (C.V.); Sabine.Gregersen@bgw-online.de (S.G.); Albert.Nienhaus@bgw-online.de (A.N.); 2Department of Work and Organizational Psychology, University of Hamburg, 20146 Hamburg, Germany; Sylvie.Vincent-Hoeper@uni-hamburg.de; 3Institute for Health Services Research in Dermatology and Nursing, University Clinic Hamburg-Eppendorf, 20246 Hamburg, Germany; a.schablon@uke.de

**Keywords:** healthcare, social services, sexual harassment, well-being, prevention and aftercare

## Abstract

Social and healthcare workers are at high risk of experiencing sexual harassment in the workplace. Although sexual harassment is detrimental to people’s well-being, only a few studies have systematically investigated social and healthcare workers’ experiences of different forms of sexually harassing behaviors by patients, clients, and residents in Germany. This study aimed to address this gap by determining the prevalence rates and frequency of nonverbal, verbal, and physical sexual harassment by patients, clients, and residents against social and healthcare workers. In addition, we examined the associations of sexual harassment with workers’ well-being and described employees’ awareness of offers of organizational support for sexual harassment prevention and aftercare. Data were collected from *n* = 901 employees working in a total of 61 facilities, including inpatient and outpatient care, psychiatric facilities, hospitals, and facilities for persons with disabilities. While the prevalence, frequency, and predominant forms of sexual harassment differed across sectors, the results indicated that nonverbal, verbal and physical sexual harassment were highly prevalent in social and healthcare work, with both men and women being affected. Furthermore, we found that sexual harassment was positively related to impaired well-being (e.g., depressiveness and psychosomatic complaints). In terms of support offers for sexual harassment prevention and aftercare, we found that approximately one-third of social and healthcare workers were not aware of any offers at their facilities. In addition to highlighting the problem of sexual harassment by patients, clients, and residents in social and healthcare settings, this study provides recommendations for the development of interventions and suggests several avenues for future research.

## 1. Introduction

Sexual harassment in the workplace occurs every day all over the world: it is a global issue that is still prevalent and taboo [1,2,3]. While the research on sexual harassment by supervisors and colleagues is substantial, knowledge is limited about sexual harassment by clients, patients, and residents [4,5,6]. Relative to other industries, the healthcare workforce is even more at risk of experiencing sexual harassment by people outside the organization (e.g., patients and clients) [7,8]. Research in the field of sexual harassment by patients, clients or residents is fragmented and fraught with problems [9], especially due to the lack of a valid assessment of this phenomenon. A major challenge is that meaningful prevalence rates are lacking. To understand sexual harassment in healthcare work, efforts must be made to draw a more comprehensive picture of the sexual harassment experienced by care workers, including that by patients, clients or residents. Although findings indicate that sexual harassment by individuals outside the organization is a prevalent issue in healthcare work [7] and that experiencing sexual harassment by patients or clients can have similar serious negative psychological consequences for the victims [10] as sexual harassment by colleagues and supervisors, empirical investigations of this phenomenon remain scarce [11]. It is therefore important to gain further evidence-based insights into this important topic. In recent years, the issue of sexual harassment at work has been very present in the media in Germany and other Western countries [12,13,14]. However, reliable and detailed, scientific knowledge is lacking about sexual harassment by patients, residents or clients [10] that describes in which sector of healthcare and social services sexual harassment is experienced, with what frequency different forms of sexual harassment are experienced and which health consequences (e.g., impaired well-being) are to be expected for the employees concerned. With such knowledge, the problem area could be described precisely, and it would be possible to derive targeted measures for the prevention and aftercare of sexual harassment in the workplace.

In mainly Western countries it is the responsibility of employers to take measures to prevent sexual harassment in the workplace and to support the individuals affected [15]. This responsibility is enshrined in law in Germany. Sexual harassment is described as unwanted, sexually determined behavior that makes a person feel uncomfortable and violates his or her dignity (§ 3 para. 4 of the General Act on Equal Treatment (AGG)). The individual assessment of the harassed person is prioritized, regardless of whether the harassing behavior is intentional or unintentional. Sexual harassment in the workplace is a multifaceted phenomenon that covers a wide range of nonverbal, verbal, and physically inappropriate sex-related behaviors at work [2,7,9,16,17]. Examples of nonverbal sexual harassment are sexualized gestures or acts, and examples of verbal sexual harassment are unwanted sexualized comments or jokes [7]. With respect to physical sexual harassment, examples include unwanted exposure and physical advances, physical coercion or compulsion to perform unwanted sexual acts and the use of blackmail to coerce the performance of sexual acts [7]. Summarizing the state of research, we postulate that scientifically based prevalence rates remain lacking on the full range of sexual harassment—namely, nonverbal, verbal and physical sexual harassment—in the sectors of healthcare and social work and only limited knowledge is available on the relationship between extra-organizational sexual harassment and indicators of employee well-being.

### 1.1. Prevalence of the Sexual Harassment of Social and Healthcare Workers by Patients, Clients, and Residents

The social and healthcare sectors in Germany are divided into the areas of healthcare (e.g., hospitals), homes (e.g., inpatient care facilities and housing facilities for disabled persons), and social services without homes (e.g., care for elderly people, people with disabilities, children and young people) [18]. Employment in the German social and healthcare sectors is dominated by women: approximately three-quarters of all employees are female [19,20].

A representative survey of 1002 employees in Germany revealed that approximately 49% of the female workforce and 56% of the male workforce have experienced sexual harassment in general in their working life [7]. Regarding general sexual harassment at work in the past three years, 13% of females and 5% of males reported being affected by sexual harassment at work [7]. According to the survey, female employees were significantly more likely than males to experience sexual harassment [7]. These findings underscore the large extent of general sexual harassment in the workplace in Germany. In an internal study of the University Hospital Charité in Berlin, Germany, 70% (76% female, 61% male) of the 737 persons surveyed stated that they had been affected by general sexual harassment at work at least once during their work in the medical field [21]. Schablon and colleagues [22] assessed general sexual harassment with a single-item measure and provided initial evidence of the varying prevalences of sexual harassment in the past 12 months in the German social and healthcare sectors, ranging from 6.4% in residential facilities for people with disabilities to 18.1% in inpatient care for elderly people. The findings underscore the concern in the social and healthcare sectors in Germany about sexual harassment in the workplace. However, it remains unclear which form of sexual harassment (nonverbal, verbal, or physical) occurs. Existing studies have often focused on verbal and physical sexual harassment while often disregarding nonverbal aspects of sexual harassment [9]. The study by Vincent-Höper and colleagues [9] provided initial evidence that all three forms of sexual harassment by patients, clients, and residents occur in social and healthcare facilities in Germany. However, in the few existing studies that have focused on the prevalence of sexual harassment by patients, clients, and residents, unvalidated scales or single-item measures have been used (e.g., “Have you been exposed to sexual harassment at your workplace during the last 12 months?”). The use of single-item measures faces various problems. For example, the answer to direct questions about sexual harassment could depend upon the respondents’ implicit definitions of sexual harassment and might result in an underestimation of prevalence rates [9]. A reason behind this might be that people are reluctant to label sexually inappropriate behaviors as sexual harassment [23] because being a victim of sexual harassment is associated with stigmatization and weakness. This reluctance may lead to those experiences being ignored or trivialized [24]. Another reason lies in the nature of healthcare work. Care workers often distinguish between intentional and unintentional sexual behaviors initiated by patients suffering dementia or other cognitive impairments [2]. Asking directly about experienced sexual harassment could imply that the actions were intentional and thus lead to an underestimation of the occurrence of sexual harassment. A single item cannot capture the different aspects of observable sexual harassment. In addition, we argue that inappropriate sexual behaviors may have an impact on care workers’ well-being, even if it is not labeled sexual harassment.

Much of the existing international research has addressed sexual harassment by colleagues and supervisors [1]. However, recent studies have revealed that in many cases of sexual harassment, customers, clients or patients are the perpetrators [7,9,21,25]. An essential characteristic of professional work in the social and healthcare sectors is the interaction with patients, clients, and residents, with the task being to support, advise and/or care for people in difficult situations. These communications are often characterized by challenging behaviors (e.g., dealing with patients with dementia and working with challenging teenagers). The ability to communicate with clients in healthcare and social services is also dependent on the degree of cognitive impairment of the individuals. In addition, caring for people requires physical proximity between the people being cared for and the social and healthcare workers. These factors could encourage the occurrence of sexual harassment by these groups. To address this gap, our study focuses on sexual harassment by patients, clients, and residents in the workplace. However, the activities and the resulting demands on workers differ depending on the type of facility (e.g., between psychiatric and ambulatory care services) in the social and healthcare sectors. Despite the known high risk of experiencing general sexual harassment in the workplace in social and healthcare services [7], there is a lack of sector-specific evidence in Germany on the prevalence and frequency of the full spectrum of sexual harassment (nonverbal, verbal, and physical) by patients, clients, and residents in various healthcare and social services.

The aim of the study is to gain insights into the different forms of observable sexually harassing behaviors (namely, nonverbal, verbal, and physical) directed at social and healthcare workers. We focus on the following research questions in our study: (1) What are the sector-specific prevalence of nonverbal, verbal, and physical sexual harassment by patients, clients, and residents in the workplace? In this context, we examine whether there are gender differences in the prevalence of nonverbal, verbal, and physical sexual harassment. Regarding the frequency of sexual harassment, our research questions are as follows: (2) Do the healthcare and social services sectors differ in terms of the frequency of nonverbal, verbal, and physical sexual harassment by patients, clients, and residents? and (3) What is the most common form of sexual harassment (nonverbal, verbal, and physical) by patients, clients, and residents in the various healthcare and social service sectors? Scientifically sound knowledge of the abovementioned questions will support employers in the fulfillment of their legal duties (e.g., the German Civil Code, the General Equal Treatment Act, and the German Occupational Health and Safety Act) [26,27,28] to initiate targeted prevention measures and provide aftercare services.

### 1.2. Consequences of Sexual Harassment at Work for Social and Healthcare Workers’ Impaired Well-Being

The World Health Organization (WHO) describes health as “a state of complete physical, mental, and social well-being and not merely the absence of disease or infirmity” (WHO, 1948). Therefore, it is important to look beyond the physical health to the mental health (e.g., impaired well-being) of employees. According to job demands-resources (JD-R) theory [29,30], job demands have an effect on employees’ impaired well-being (e.g., emotional exhaustion, depression, etc.). Job demands are defined as physical, social or organizational job aspects that require sustained physical and/or mental effort and are therefore associated with certain psychological/or physiological costs [31]. Job demands may become burdensome when they exceed employees’ capabilities [29,30]. An example is emotionally demanding interactions with clients or customers [30]. Sexual harassment in the workplace, as an unwanted sex-related behavior, is a job demand that exceeds sufferers’ resources and capabilities and may threaten their well-being [32]. A meta-analysis of the antecedents and consequences of sexual harassment at work revealed significant effects of sexual harassment on physical health and impaired well-being [11]. The small number of studies on sexual harassment by clients and customers indicated severe detrimental effects on employees’ well-being and indicators of mood [2,8,10,25,33,34,35]. However, most studies have examined sexual harassment in general. To obtain a more nuanced picture of the relationship between experiences of sexual harassment in the workplace and employee well-being, it is necessary to consider different forms of sexual harassment. The purpose of this study is to use a valid instrument to examine the differential effects of observable nonverbal, verbal, and physical sexual harassment on well-being.

A recent study by Vincent-Höper and colleagues [9] on sexual harassment by clients, patients and residents against 305 German healthcare and social services employees found substantial effects between nonverbal, verbal, and physical sexual harassment and indicators of impaired mental health (e.g., emotional exhaustion, stress, depressiveness and psychosomatic complaints). This study provided preliminary evidence regarding the relevance of the three forms of sexual harassment by patients, clients, and residents for healthcare and social workers impaired well-being. Nevertheless, more research based on a large amount of data is required to replicate the observed associations of nonverbal, verbal and physical sexual harassment by patients, clients, and residents against employees in healthcare and social services. Our study is based on a large amount of data and allows us to verify and corroborate the study results of Vincent-Höper and colleagues [9]. Summarizing the aforementioned findings, we hypothesize (*H1*) that all forms of sexual harassment (nonverbal, verbal, and physical) by patients, clients, and residents against healthcare and social services workers show substantial correlations with workers’ impaired well-being.

### 1.3. Awareness of Support Offers for Sexual Harassment Prevention and Aftercare at Work

There are a number of measures for sexual harassment prevention and aftercare at work. Examples include company protection policies for dealing with sexual harassment, the establishment of a complaint office in accordance with the AGG, training in dealing with sexual harassment, and internal team case discussions. However, it is unclear how well known and widespread these support offers are in healthcare and social services organizations. Therefore, this study additionally examines which offers for sexual harassment prevention or aftercare are known in healthcare and social services institutions.

## 2. Materials and Methods

### 2.1. Ethics Approval and Consent to Participate

This research study received research ethics committee approval from the Local Ethics Committee of the Faculty of Psychology and Human Movement at the University of Hamburg (no. 2020_331). All participants were informed about the purpose of the study, the voluntary nature of participation, data privacy and anonymity. As employees’ participation in the study was anonymous, no conclusions can be drawn about individuals or organizations. Due to the potentially upsetting subject matter of this study, the study participants were given the name of a consultation office close to their homes as well as the contact information for an anonymous telephone counseling service on the topic of sexual harassment/abuse. Consent to participate in the study was given in writing by the participating organizations.

### 2.2. Procedure and Sample

A set of 358 organizations in the healthcare and social services sector (inpatient and outpatient care facilities, psychiatric facilities, hospitals and rehabilitation hospitals, and workshops and housing facilities for disabled persons) located in four federal states in Germany was drawn at random from a dataset of insured organizations from the German Social Accident Insurance Institution for Health and Welfare Services (BGW). The recruitment of study participants (employees with regular contact with people in need of care) from the random sample was done in writing and by telephone by the BGW. Sixty-six organizations agreed to participate in the study and distribute the paper questionnaires to their employees. A total of 5970 questionnaires were sent to these companies. The survey period was from July 2019 to January 2020. A total of 929 questionnaires (response rate: 16%) were returned from 60 participating organizations. Of the 929 questionnaires, 901 were usable. To ensure high-quality data entry, a random sample of 10% of the questionnaires was entered a second time and examined for discrepancies in the data entry with the overall sample. Any discrepancies were checked manually and corrected if necessary. Overall, the quality of the data entry was satisfactory. This approach yielded a final sample size of *n* = 901 employees (see Table 1).

The average age of the study participants was 42.62 years (SD = 12.84, range: 17–75), and the average number of working hours per week was 32.62 (SD = 8.26, range: 6–60). Due to the small number of participants from the rehabilitation hospital sector, the hospital and rehabilitation hospital sectors are combined and referred to as “hospitals” in the following.

### 2.3. Measures

Sexual harassment (including nonverbal, verbal, and physical sexual harassment) during the past 12 months in the form of observable behaviors by patients, clients or residents was measured with 14 items from the Sexually Harassing Behavior Questionnaire (SHBQ-X) from Vincent-Höper et al. [9]. The items were scored on a six-point Likert-type scale (1 = “never”, 2 = “once in 12 months”, 3 = “every few months”, 4 = “every few weeks”, 5 = “every few days”, or 6 = “(nearly) every day”). For item wording see Table 2.

Additionally, we asked about attempted rape (“I have experienced attempted rape” [9]) and forced sexual acts (“I was forced to perform sexual acts” [9]) at work, as extreme forms of physical sexual harassment, with one item each (response format “yes”/“no”).

As indicators of impaired well-being, we measured emotional exhaustion with seven items from the German version [36] of the Maslach Burnout Inventory [MBI General Survey; [37]. A sample item is “I feel drained from my work”. The responses were provided on a six-point Likert-type scale (1 = “never” to 6 = “several times a week”). As a second indicator of impaired well-being, we assessed depressiveness in the nonclinical context with eight items [38] scored on a seven-point Likert-type scale (1 = “never” to 7 = “almost always”). A sample item is “I have sad moods”. Third, psychosomatic complaints were measured with six items from the Psychosomatic Complaints in the Nonclinical Context scale from Mohr and Müller [38]. The responses were provided on a five-point Likert-type scale (1 = “never” to 5 = “almost daily”). A sample item is “Do you have headaches?” To assess the participants’ awareness of different support offers for the prevention and rehabilitation of sexual harassment, we asked, “Which of the following sexual harassment prevention or aftercare services are you aware of at your organization? (multiple answers possible)” (examples of support offers are company protection policies, complaint office in accordance with the AGG, training, and internal team case discussions).

### 2.4. Statistical Analyses

The statistical analyses were performed in IBM SPSS Statistics version 25 [39] and R version 4.0.2. [40]. To test the factor structure of the sexual harassment measure, confirmatory factor analysis (CFA) using the lavaan package [41] was conducted. We used the robust maximum likelihood (MLR) with robust (Huber-White) standard errors and a scaled test statistic that is (asymptotically) equal to the Yuan-Bentler test statistic [42], and we used the full information maximum likelihood (FIML) estimator for parameter estimation. The fit of the model to the data was assessed using the robust scaled chi-square value (χ^2^), the robust comparative fit index (CFI) as a goodness-of-fit index, and the robust root mean square error of approximation (RMSEA) and robust standardized root mean square residual (SRMR) as badness-of-fit indices. General guidelines suggest that CFI values close to 0.95 or higher, SRMR values of 0.08 or lower, and RMSEA values of 0.06 or lower indicate adequate fit [43,44]. A nonsignificant chi-square indicates good model fit [44]. Furthermore, descriptive statistics such as the frequency, mean, standard deviation, and range, as well as reliability and bivariate Pearson correlations were calculated. If all items in a sexual harassment subscale were answered “never”, the participant was considered to have not experienced this form of sexual harassment in the past 12 months. We report the occurrence of sexual harassment by calculating both the prevalence rates and the mean values of the sexual harassment scales to demonstrate the extent of sexual harassment in more detail based on its frequency. If at least one item in a sexual harassment subscale was answered with at least “once in 12 months”, the participant was considered to have experienced this form of sexual harassment in the past 12 months. Dichotomizing the variables yielded an understandable percentage of employees who experienced sexual harassment at work (at least one experience in the last 12 months) hence, providing information on the mere prevalence. For the further analyses regarding the extent of the sexual harassment at work, the more scientifically exact measure of mean values was used. These also allowed us to conduct inferential statistical methods such as t-tests and correlations with the indicators of impaired well-being. For mean comparisons, t-tests were performed, including testing for variance homogeneity. We compared the mean value of one sector with the mean value of the remaining sectors. Due to anonymization, no data analysis at the team or organizational level was possible.

## 3. Results

To make the subscales uniform and to be able to report comparable prevalences, the verbal sexual harassment subscale was reduced from the original six items to four items. Otherwise, the measured behavior of the verbal sexual harassment would be overrepresented in a comparison, leading to biased results with regard to the prevalences. When conducting organizational research, survey length is at a premium, and using comprehensive, psychometrically sound measures is critical [45]. To develop a valid, uniform, and more parsimonious version of the SHBQ-X, we reduced the number of items based on recommendations from Stanton and colleagues [45], who suggest selecting items based on three criteria: judgmental qualities (e.g., subjective judgment of face validity and other nonstatistical considerations), internal qualities (e.g., item qualities in reference to the scale), and external qualities (e.g., relations with meaningful external criteria). Therefore, we deleted two items based on the judgment of four experts, the discriminatory power of the items, factor loadings (CFA), and correlations with indicators of employee well-being. This approach resulted in a more balanced measure of sexually harassing behavior, including three scales with four items each. In the CFA, the shortened scale, which included verbal sexual harassment items 3 to 6 (see Table 2), showed a very good fit to the data (χ^2^ = 56.962, df = 51, *p* = 0.263; scaling correction factor = 7.241; robust CFI = 0.995; robust RMSEA = 0.031, robust RMSEA CI 0.000–0.067; robust SRMR = 0.031). The scales demonstrated acceptable reliabilities (Cronbach’s alpha: 0.80 to 0.92).

### 3.1. Prevalence of Sexual Harassment in Healthcare and Social Services

The sector- and gender-specific prevalence rates of experienced nonverbal, verbal, and physical sexual harassment at work in the past 12 months are shown in Table 3.

The results show that 62.5% of all respondents had experienced nonverbal sexual harassment, 67.1% had experienced verbal harassment and 48.9% had experienced physical sexual harassment by patients, clients or residents in the workplace. With respect to gender, 69.7% of the male respondents and 60.7% of the female respondents had experienced nonverbal sexual harassment. A total of 59.1% of the male participants and 69.2% of the female participants reported experiencing verbal sexual harassment. The prevalence of physical sexual harassment was 41.7% for the men and 50.7% for the women who responded. The gender differences in the prevalence rates were significant (*p* = 0.01/*p* = 0.03).

The prevalence of nonverbal, verbal, and physical sexual harassment in the workplace in the past 12 months varied across the healthcare and social service sectors. The prevalence of nonverbal sexual harassment ranged from 48.1% in outpatient care services to 73.6% in workshops for disabled persons. Regarding verbal sexual harassment, housing facilities for disabled persons had the lowest prevalence rate (57.7%), while 75.9% of the study participants from hospitals had experienced verbal sexual harassment in the past 12 months. The prevalence of physical sexual harassment was the lowest in psychiatric facilities (38.0%), and the highest rate was reported by staff in inpatient care facilities (53.0%). Beyond nonverbal, verbal, and physical sexual harassment, 1.6% (15 individuals) had experienced attempted rape, and 0.9% (9 individuals) had been forced to perform sexual acts at work during their working lives.

### 3.2. Sector Comparison—Frequency of Sexual Harassment

In the total sample, the mean scores for nonverbal sexual harassment (M = 1.79, SD = 0.96), verbal sexual harassment (M = 1.97, SD = 1.08), and physical sexual harassment (M = 1.55, SD = 0.87) differed significantly (*p* < 0.001) from each other. The patterns of nonverbal, verbal and physical sexual harassment differed among the different sectors (see Figure 1).

The employees in the workshops for disabled persons sector experienced nonverbal sexual harassment significantly more often than the average of the employees in the other sectors. In outpatient care services and hospitals, the average nonverbal sexual harassment values were significantly lower than those in the other sectors. In inpatient care facilities and outpatient care services, verbal sexual harassment was significantly more frequent than in the other sectors. There was no significant difference in the mean verbal sexual harassment score between inpatient care facilities and outpatient care services. The employees in the workshops for disabled persons sector experienced verbal sexual harassment significantly less often than the employees in the other sectors. Physical sexual harassment was significantly lower in psychiatric facilities and hospitals than in the other sectors.

### 3.3. Sector-Specific Patterns—Frequency of Nonverbal, Verbal, and Physical Sexual Harassment

The predominant form of sexual harassment (nonverbal, verbal, or physical) varied by sector. In the workshops for disabled persons sector, nonverbal sexual harassment was significantly more frequent (*p* < 0.001) than the other forms of sexual harassment. This pattern was only seen in the workshop sector and therefore not in housing facilities for disabled persons. In the care sectors (inpatient care facilities and outpatient care services) and in hospitals, verbal sexual harassment was significantly more frequent (*p* < 0.001) than the other forms of sexual harassment. In inpatient care facilities, verbal sexual harassment was experienced more frequently than nonverbal sexual harassment, and nonverbal sexual harassment was experienced more frequently than physical sexual harassment (*p* < 0.001/*p* < 0.004). A ranking could thus be derived. In psychiatry and housing facilities for disabled persons, verbal and nonverbal sexual harassment occurred significantly more often than physical sexual harassment (*p* < 0.001/*p* < 0.008).

### 3.4. Associations between Sexual Harassment and Impaired Well-Being

Table 4 shows the correlations between the three forms of sexual harassment (nonverbal, verbal, and physical) against social and healthcare workers by patients, clients, or residents and the social and healthcare workers’ impaired well-being (emotional exhaustion, depressiveness, and psychosomatic complaints).

All three forms of sexual harassment showed substantial significant positive correlations, at relatively similar levels, with the impaired well-being of the individuals affected. The correlations with indicators of impaired well-being ranged from *r* = 0.13 to *r* = 0.22 for nonverbal, *r* = 0.21 to *r* = 0.28 for verbal, and *r* = 0.17 to *r* = 0.25 for physical sexual harassment. The correlations of all forms of sexual harassment were found to be minimally stronger with emotional exhaustion (*r* = 0.22 to *r* = 0.28) than with depressiveness and psychosomatic complaints (*r* = 0.13 to *r* = 0.21/*r* = 0.13 to *r* = 0.25). Therefore, our hypothesis H1 was confirmed.

### 3.5. Awareness of Support Offers for the Prevention and Rehabilitation of Sexual Harassment at Work

Table 5 displays the results on the participants’ awareness of support offers for prevention and after care of sexual harassment in social and healthcare institutions.

The level of awareness of the various support offers varied (see Table 5). Of the study participants, 6.8% reported that briefings were offered, 30.6% cited case reviews/supervision as measures in their organization, 32.5% were not aware of any measures, and 3.8% stated that other measures were offered at their facilities.

As “other“, employee representatives, women’s representatives, and confidants or managers were mentioned as contact persons for prevention or aftercare following sexual harassment, as well as working groups on the topic and consultation through Pro Familia. It was also reported that, on request, there is no longer any outreach to the involved patients; another person in house takes care of the clients concerned, and a “We won’t leave you alone” climate is created. One participant stated that after an incident of sexual harassment, the process was discussed in the team meeting and management meeting across divisions, training was formulated, and the process was made available on the intranet. The development of a pedagogical approach was also mentioned as a measure to prevent sexual harassment.

## 4. Discussion

In a large sample of 901 social and health workers, we demonstrated high prevalence and frequency of nonverbal, verbal, and physical sexual harassment by patients, clients, and residents. In contrast, we observed a low level of awareness of support services for sexual harassment in the social and healthcare sectors in Germany.

A high proportion, namely, just under half to approximately two-thirds of the participating social and healthcare workers, had experienced nonverbal, verbal, and/or physical sexual harassment by patients, clients, or residents in the past 12 months in their work. Our findings are consistent with those of Schröttle and colleagues [7], who reported a high risk of experiencing general sexual harassment in the workplace in health and social care. Compared with the prevalence of general sexual harassment (9%) toward professionals in Germany in the past three years [7] and the prevalence of general sexual harassment of social and health care workers in the past 12 months in Germany [22], the prevalence rates we found in this study of nonverbal, verbal, and physical sexual harassment by patients, clients, and residents alone in the past 12 months in the social and health care sector can be considered very high, but are in line with previous research [2,21,23].

However, our results also show that the overall figures for healthcare and social services mask significant sector-specific differences. All three forms of sexual harassment—nonverbal, verbal, and physical—occur to varying degrees in healthcare and social services. The highest prevalence rate for nonverbal sexual harassment was found in workshops for disabled persons, and the lowest prevalence rate was found in outpatient care services. Regarding verbal sexual harassment, the highest prevalence rate was reported by healthcare workers in hospitals, and the lowest prevalence was reported in housing facilities for disabled persons. The lowest prevalence rate for physical sexual harassment was reported among employees from psychiatric facilities, and the highest prevalence rate was reported by staff in inpatient care facilities. With regard to health care and social services, it is therefore reasonable to look at the areas in such a differentiated way. This suggestion is consistent with Schablon and colleagues’ [22] findings on general sexual harassment, which also showed sector-specific differences in prevalence rates. Moreover, we found that nonverbal sexual harassment was experienced significantly more often by male workers, but that verbal and physical sexual harassment was experienced significantly more often by female social and healthcare workers. According to the study by Schröttle and colleagues [7], female employees are significantly more likely to experience sexual harassment than male employees. In this respect, the gender difference in the prevalence of nonverbal sexual harassment by patients, clients, and residents in the workplace observed in this study is surprising. One reason could be that previous studies often asked directly about (general) sexual harassment by general harassers; therefore, the prevalence of nonverbal sexual harassment against men may have previously been underestimated [3].

In addition to differences in prevalence rates, the study also showed sector-specific differences in the frequency with which the social and healthcare workers experienced different forms of sexual harassment by patients, clients, and residents. In workshops for disabled persons, nonverbal sexual harassment against employees occurred more frequently than in the other sectors. Moreover, nonverbal harassment was still the most common form of sexual harassment in workshops for disabled persons. Verbal sexual harassment in outpatient care services and inpatient care facilities was more prevalent than both verbal sexual harassment in the other sectors and the other forms of sexual harassment in these two sectors. In hospitals, verbal sexual harassment was the predominant form of sexual harassment but was no more common than in other sectors. With regard to physical sexual harassment, no sector was affected more frequently than average. Workers in care (inpatient and outpatient) and disability care (workshops and housing facilities for disabled persons) experience physical harassment on average frequently. In psychiatry and housing facilities for disabled persons, physical sexual harassment occurred less often than verbal and nonverbal sexual harassment. However, there were sectors that were less likely to be affected by certain forms of sexual harassment than others. In inpatient care facilities, it was even possible to rank the frequency of the forms of sexual harassment (1. verbal, 2. nonverbal, and 3. physical), with significant differences in the frequency of each form. In fact, our results showed substantial significant differences in the frequency of nonverbal, verbal, and physical sexual harassment in social and healthcare settings. The reasons for these differences could be the different situational factors [11] and working conditions of the individual sectors as well as the different restrictions (e.g., dementia, cognitive impairments, etc.) and challenges of the clientele. The above findings on the prevalence and frequency of sexual harassment indicate the need for a differentiated, sector-specific examination of all three forms of sexual harassment in the social and health care sectors and a consideration of both female and male sufferers.

In our study, we also examined the relations between different forms of sexual harassment against social and healthcare workers by patients, clients, and residents and their impaired well-being. In line with our hypothesis (*H1*), we found substantial correlations between all forms of sexual harassment and social and healthcare workers’ impaired well-being (emotional exhaustion, depressiveness and psychosomatic complaints). Previous research shows that, due to the large number of predictors of impaired well-being, correlations of 0.20 to a maximum of 0.30 between job demands and impaired well-being are to be expected [46]. Thus, all three forms of sexual harassment represent relevant job demands in view of social and healthcare workers’ well-being. Nonverbal, verbal, and physical sexual harassment by patients, clients, and residents may exceed social and healthcare workers’ capabilities and threaten their well-being [29,30]. The strength and direction of the associations between the three forms of sexual harassment and social and healthcare workers’ well-being found in our study are similar to those found in the study by Vincent-Höper and colleagues [9]. Furthermore, our results match previous research findings on the effect of general sexual harassment on employees’ well-being and indicators of mood [2,8,10,25,33,34,35]. Based on a large amount of data, our study has certainly provided an even more nuanced picture of the relations between specific forms of sexual harassment and impaired well-being. Our study is the first to provide scientifically based, differentiated evidence on the sector-specific prevalence and frequency of sexual harassment of social and healthcare workers by patients, clients, and residents and the associations of sexual harassment with the impaired well-being of sufferers. We used validated measures to capture the full range of inappropriate nonverbal, verbal, and physical sexual behaviors instead of using general single-item measures or direct questions about sexual harassment.

This study provides insights into awareness of support services for the prevention and rehabilitation of sexual harassment in social and healthcare institutions. The results of our study show that the level of awareness of various support offers varies. Almost one-third of the respondents were not aware of any measures. It is possible that some services are appropriate for the prevention of general violence but not for the prevention of sexual harassment (e.g., de-escalation training). Surprisingly, many respondents were aware of de-escalation training related to sexual harassment. Further research would be necessary to determine the type of de-escalation training offerings that employees are aware of. In addition, the results of our study suggest that more education on the issue of sexual harassment in social and health care settings in the context of the employer’s duty of care is needed. Additional efforts should be made to provide information about the meaning and consequences of sexual harassment in the workplace, as well as to develop appropriate sexual harassment prevention and aftercare measures and to provide assistance for their implementation in social and healthcare settings. The need for more education on sexual harassment in the healthcare and social services sectors (e.g., meaning and consequences and prevention and aftercare measures) and intensified assistance in implementing workplace interventions (prevention and aftercare) may be transferable to other countries.

### 4.1. Future Directions

In terms of future research, it would be useful to develop a comprehensive theoretical framework of sexual harassment by patients, clients, and residents. This framework should include antecedents, consequences, and boundary conditions that explain the dynamic process of sexual harassment. Considering that we conceptualized sexual harassment as a job demand that may impair employee well-being (e.g., burnout and depression), the JD-R model [29,30] would offer a suitable theoretical basis for the development of a conceptual framework of sexual harassment. Expanding the JD-R model by including antecedents of sexual harassment may help identify potential risk and protective factors for the occurrence of sexual harassment by patients, clients, and residents.

A comprehensive framework of sexual harassment should take a multilevel perspective and include risk and protective factors at both the organizational and individual levels. Meta-analytical findings provide support that the organizational climate in terms of the organization’s intolerance for sexual harassment and explicit norms and policies regarding sexual harassment are key factors in the occurrence of sexual harassment [11]. Future studies should aim to identify organizational risk factors that increase the likelihood of experiencing sexual harassment by patients, clients, and residents, such as unfavorable working conditions (e.g., avoiding working alone, lacking qualifications, and having insufficient information), power distance, job gender context, organizational climate, and the patient-staff ratio. In addition, we encourage future research to explore personal risk factors for employees (e.g., neuroticism and low self-esteem) and patients, clients, and residents (e.g., gender; age; hostility; narcissism; and specific diseases, such as dementia). In this context, the underlying reasons for the different prevalences of sexual harassment between men and women should also be explored. Moreover, the prevalence of sexual harassment in the workplace of healthcare workers depends on whether the occupational training has been completed or the person is still in training [47]. Further research should investigate the prevalences of sexual harassment at work for additional subgroups.

Given the frequency of sexually harassing behaviors toward healthcare workers and their severe negative effects on care workers’ mental health [10,34], another important direction for future research is the identification of resources to help healthcare workers cope with sexual harassment by patients and clients. Building on the JD-R model, we suggest that several organizational resources (e.g., social support and organizational climate) and personal resources (e.g., hardiness and self-efficacy) may buffer the negative association between sexual harassment by patients/clients and social and healthcare workers’ well-being. Longitudinal research is needed to differentially examine the causal relationships between nonverbal, verbal, and physical sexual harassment and indicators of health and well-being. Future research should use mixed methods approaches (e.g., surveys, interviews, and observations), different study designs (e.g., a multilevel design), and multiple indicators of employee well-being (e.g., self-reports and behavioral and physiological measures) to draw more definitive conclusions [48,49].

Moreover, the findings on the substantial associations between sexual harassment and social and healthcare workers’ mental health in this study emphasize the importance of developing interventions that prevent the incidence or alleviate the adverse effects of sexual harassment. Because organizations will not be able to prevent all incidents of sexual harassment, knowledge is needed on how to prevent or reduce workers’ stress reactions after an assault occurs. According to meta-analytical findings [50,51], one focus for interventions should be on the structural conditions of organizations. Further research should also investigate the reasons that prevent or support the implementation of measures against sexual harassment in companies. It should also be examined whether there is a need for sector- and gender-specific measures for the prevention and aftercare of nonverbal, verbal and physical sexual harassment or whether the existing support offers already accomplish these aims.

Future studies should investigate the specific content of postevent intervention strategies. Qualitative methods may help identify employees’ needs after experiences of sexual harassment and develop tailored postevent interventions see [52]. To give concrete and evidence-based recommendations for managing sexual harassment, future research should conduct high-quality intervention studies (e.g., randomized controlled trials) to rigorously evaluate the effectiveness of different offers for assistance after exposure to sexual harassment.

Although this study investigated the frequency of different sexually harassing behaviors, social and healthcare workers’ interpretation of the specific behaviors in terms of their intensity and harmfulness may have important implications for their reactions. For example, the interpretation of a behavior as an intentional act of harm may strengthen the negative relationship between the experience of sexual harassing behaviors perpetrated by patients, clients, or residents and employee well-being. In contrast, social and healthcare workers’ perceptions that sexually harassing behaviors are due to patients’ disease patterns (e.g., dementia) or medication might attenuate the negative relationship between sexually harassing behaviors and well-being. We suggest that using qualitative methods to explore social and healthcare workers’ attributions of the causes of sexually harassing behaviors advances the understanding of moderators of the relationship between sexually harassing behaviors and well-being.

In this study, we took an extra-organizational perspective of sexual harassment. Previous research has shown that intra-organizational sexual harassment by colleagues and supervisors has adverse effects on well-being [1]. Future research may simultaneously examine different perpetrators of harassment (e.g., colleagues, supervisors, clients, etc.) to identify differential effects and ways to decrease the occurrence and health-impairing impact of sexual harassment.

### 4.2. Limitations

Some limitations should be considered in the interpretation of the findings. Our study collected a large amount of data. However, the sample was not representative, which means that the generalization of our study results to all employees in the sectors examined in this study should be performed with caution. Due to very country-specific frameworks of sexual harassment in the workplace, such as those regarding labor laws and working conditions in healthcare and social services, the generalizability of the findings and implications to other countries and contexts is limited. Moreover, we were able to achieve a response rate of only 16% during data collection, which may have been due to the taboo and stigma surrounding the subject of this study. Therefore, it is possible that the study results were influenced by selection bias. For example, our study results could have been biased by nonrandom differences between individuals who were and were not willing to participate (e.g., experience of sexual harassment in the workplace). Our study used cross-sectional data, which restricts any conclusions about the causality of the effects. Moreover, the analyses used single-source, single-method data. The participants assessed both their experience of sexual harassment at work and their well-being. Therefore, common method variance may have inflated the relationships [53,54].

### 4.3. Practical Implications

Sexual harassment and violence in the workplace are often underestimated, especially in the healthcare and social services sectors. However, our study shows that the risk of experiencing sexual harassment and violence at work is particularly high in these sectors, which is in line with previous findings [7]. Our results show that both women and men are affected and that there are clear sector-specific differences. In addition to physical sexual harassment, nonverbal and verbal sexual harassment by patients, clients, and residents in facilities need to be addressed in a differentiated manner.

Crossing personal or intimate boundaries is part of employees’ everyday lives in social and health care, as it is often necessary to ensure good treatment and care. This aspect of social and healthcare work makes it all the more important to draw a clear line between boundary violations and sexual harassment in facilities. Many forms of sexual harassment are interpreted as such by only approximately two-thirds of those affected [55]. Therefore, it is particularly important to ask about concrete observable behaviors of sexual harassment and not only about individual interpretations. The differentiation of nonverbal, verbal and physical forms of sexual harassment provides a good orientation for the development of measures in social and healthcare institutions.

The development of measures should pursue three goals: create awareness of the issue (Watch), introduce aftercare services (Protect), and propose preventive measures (Prevent). Prevention measures and offers of protection and care should be addressed equally to women and men and differentiated according to the type of sexual harassment and sector. The present study has shown that employees are often unaware of offers for the prevention and aftercare of sexual harassment and violence in the workplace. The importance of the implementation of a management concept has been emphasized by work from Schablon and colleagues [22]. Their findings on general violence in the workplace have shown that employees feel less stressed if the company has established violence management concepts [22]. We also recommend establishing a comprehensive management concept (that considers the three levels: Watch, Protect, and Prevent) in health and social care facilities for the prevention and aftercare of sexual harassment in the workplace. The concept should be based on a clear position of the company against sexual harassment, including mission statements and company or service agreements on dealing with sexual harassment that clearly define sexual harassment and make the various forms clear (create awareness of the issue—Watch). The enforcement and review of agreements and procedural rules must also be clarified (e.g., emergency plan and aftercare). A complaint procedure should be established and specifically regulated. Such a procedure should include topics such as the protection of trust and anonymity, neutrality and independence from instructions of the complaint office, and the documentation of complaints (introduce protection offers—Protect). Training to raise awareness of the issue of sexual harassment, the design of work processes, and regular and mandatory training are also essential for the prevention of sexual harassment in the workplace (preventive measures—Prevent). Rights, obligations and action strategies should be actively communicated within facilities and accepted by all stakeholders (employees, sufferers, employers, leaders and employee representatives).

## 5. Conclusions

Sexual harassment against social and healthcare workers by patients, clients, and residents is a common phenomenon. All forms of sexual harassment (nonverbal, verbal, and physical) by patients, clients, and residents occur with varying prevalences and frequencies in both genders and in all social and healthcare sectors studied. All forms of sexual harassment show substantial associations with the impaired well-being of sufferers. We therefore conclude that the management of sexual harassment in the social and healthcare sectors in Germany requires a specific and differentiated approach. A sector-specific, gender-specific, harasser-specific consideration and differentiated measurement of all three forms of observable inappropriate sexual behaviors is an important prerequisite for obtaining differentiated and valid findings. This form of investigation helps to derive and improve problem-oriented measures for the prevention and aftercare of sexual harassment.

## Figures and Tables

**Figure 1 ijerph-18-05198-f001:**
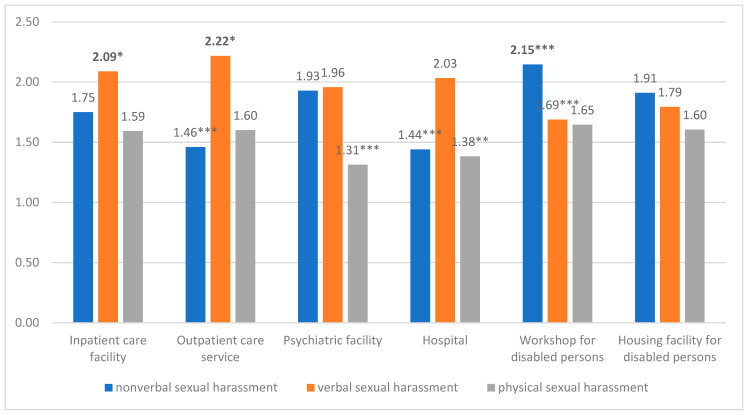
Comparison of the mean values of sexual harassment across the healthcare and social services sectors. Note: *n* = 901. Mean values that are greater than the mean value of the group of other sectors are marked in bold. Sexual harassment of social and healthcare workers by patients, clients, or residents is depicted. * *p* < 0.05, ** *p* < 0.01, *** *p* < 0.001.

**Table 1 ijerph-18-05198-t001:** Sample description.

	Count	In %
Sex (*n* = 900)		
Male	183	20.3
Female	717	79.6
Age (in years) (*n* = 889)		
<29	194	21.8
30 to 39	181	20.4
40 to 49	178	20.0
50 to 59	265	29.8
>60	71	8.0
Contact with patients/clients/residents (in years) (*n* = 899)		
>1 year	869	96.4
<1 year	30	3.3
Sector (*n* = 901)		
Inpatient care facility	292	32.4
Outpatient care service	107	11.9
Psychiatric facility	81	9.0
Hospital	115	12.7
Rehabilitation hospital	8	0.9
Workshop for disabled persons	168	18.6
Housing facility for disabled persons	130	14.4
Working hours per week (*n* = 878)		
<19	52	5.9
20 to 29	184	21.0
30 to 39	494	56.2
>40	148	16.9
Qualification (*n* = 889)		
Examined nurse	385	42.7
Care assistant	98	10.9
Employee without nursing education	39	4.3
Social pedagogue/social worker	53	5.9
Curative educator	57	6.3
Physician	12	1.3
Apprentice	32	3.6
Intern/Federal Voluntary Service/person performing year-long voluntary service	7	0.8
Group leader in a workshop for disabled persons	70	7.8
Other profession/activity	136	15.1
Leadership function (*n* = 879)		
Yes	189	21.0
No	690	76.6

Note: *n* = 901.

**Table 2 ijerph-18-05198-t002:** Items of the Sexually Harassing Behavior Questionnaire (SHBQ-X [7]).

Nonverbal Sexual Harassment	Verbal Sexual Harassment	Physical Sexual Harassment
I have witnessed sexual acts (e.g., masturbation) I have witnessed sexual gestures Someone has unnecessarily exposed themselves in front of me I have witnessed sexual harassment/violence among patients/clients/residents	I have been whistled at I have received repeated requests for dates I have been sexually complimented I have been told suggestive/offensive stories or jokes I have been exposed to verbal sexual innuendo I have been asked intrusive or personal questions by a client (e.g., requests for body measurements, relationship status, or sexual preferences)	I have been hugged in a way that made me feel uncomfortable I have been petted or patted in a way that made me feel uncomfortable I have been touched in a way that made me feel uncomfortable I have been kissed in a way that made me feel uncomfortable

**Table 3 ijerph-18-05198-t003:** Prevalence of sexual harassment in the healthcare and social services sectors.

Sector	Nonverbal Sexual Harassment		Verbal Sexual Harassment		Physical Sexual Harassment	
	Count	In %	Count	In %	Count	In %
Inpatient care facility (*n* = 292)	181	62.5	198	69.0	153	53.0
Outpatient care service (*n* = 107)	50	48.1	74	70.5	53	50.5
Psychiatric facility (*n* = 81)	53	66.2	54	68.3	30	38.0
Hospital (*n* = 123)	61	50.3	92	75.9	57	47.0
Workshop for disabled persons (*n* = 168)	123	73.6	103	62.0	77	46.9
Housing facility for disabled persons (*n* = 130)	89	69.0	74	57.7	63	49.5
Total (*n* = 884–890; 177–182 ^a^/706–707 ^b^)	557 (127 ^a^/430 ^b^)	62.5 (69.7 ^a/^60.7 ^b^)	595 (106 ^a/^489 ^b^)	67.1 (59.1 ^a/^69.2 ^b^)	433 (74 ^a/^359 ^b^)	48.9 (41.7 ^a^/50.7 ^b^)

Note: *n* = 901 (183 ^a^/717 ^b^) including missing variables. ^a^ = male, ^b^ = female; sexual harassment of social and healthcare workers by patients, clients, or residents is depicted.

**Table 4 ijerph-18-05198-t004:** Correlations between sexual harassment and impaired well-being.

	Verbal Sexual Harassment	Physical Sexual Harassment	Emotional Exhaustion	Depressiveness	Psychosomatic Complaints
Nonverbal sexual harassment	0.46 ***	0.50 ***	0.22 ***	0.13 ***	0.13 ***
Verbal sexual harassment		0.52 ***	0.28 ***	0.21 ***	0.25 ***
Physical sexual harassment			0.25 ***	0.17 ***	0.18 ***
Emotional exhaustion				0.60 ***	0.64 ***
Depressiveness					0.64 ***

Note: *n* = 901. Sexual harassment of social and healthcare workers by patients, clients, or residents is depicted. *** *p* < 0.001.

**Table 5 ijerph-18-05198-t005:** Awareness of support offers for the prevention and rehabilitation of sexual harassment in social and healthcare institutions.

Support Offers for the Prevention and Rehabilitation of Sexual Harassment	Count	In %
Guidelines/protection policy/company agreement	221	24.5
Further education/training courses	264	29.3
Complaint office in accordance with the AGG	111	12.3
Informational material	153	17.0
Topic addressed in vocational training	160	17.8
Topic addressed in case discussions/supervision	276	30.6
De-escalation training with regard to sexual assault/aspects	161	17.9
In-house support meetings by social services/pastoral care, etc.	149	16.5
Instructions/briefing	61	6.8
Other	34	3.8
No measures known	293	32.5

Note: *n* = 901. Multiple answers possible.

## Data Availability

The data are available from the authors upon request.

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
