# Peer review of "Sexual Harassment by Patients, Clients, and Residents: Investigating Its Prevalence, Frequency and Associations with Impaired Well-Being among Social and Healthcare Workers in Germany"

_ijerph, 2021, doi:10.3390/ijerph18105198_

Round 1

Reviewer 1 Report

This study is the first in its field that provides scientifically based evidence on sector-specific prevalence and frequency of sexual harassment (non-verbal, verbal, and physical) of social and healthcare workers by patients, clients, and residents and its corresponding associations with the well-being (emotional exhaustion, depression, psychosomatic symptoms) of affected individuals. This study further provided insight into workers' awareness of support services for the prevention and rehabilitation of sexual harassment in their respective institutions.

The article is well-organized, and contains all expected components (abstract, introduction, methods, etc.). Each section is well-developed, and broken into sub-sections as appropriate, resulting in a paper that is easy to follow from the reader's point of view. The authors do a good job of synthesizing the existing literature on the topic, summarizing results previously found, and explaining how their study will specifically differ and add value to this field. The methodology is clearly explained; the authors used validated scales and measurements in their study, as well as sound statistical analysis of the data. This study provided foundational scientifically sound data now that opens many doors for further research in the field that may contribute ultimately to minimizing sexual harassment and alleviating its effects in the healthcare sector.

The limitations of the study are also well described.

Overall well-written and easy to understand, however there are some areas that may benefit from clarification, specifically the following suggestions are made:

  • In line 145, there is a comment regarding the legal duty of the employers, and this may be strengthened by providing specific statute.
  • In line 220, further clarification needed on types of "discrepancies" examined.
  • In line 602, it is unclear what "unwanted boundary violations" are as the phrase implicitly suggests there are such things as "desired boundary violations."

Reviewer 2 Report

I would like to thank the authors for their efforts to zoom into the subject matter with a specific focus on Germany. The title of the study captures the thematic focus. The importance of the study and the soundness of the proposed hypotheses are supported by the evidence included in the study. Sufficient details are provided to replicate the proposed experimental procedures and analysis. The section of limitations and future directions excellently addresses potential questions regarding the return rate of the survey and recommendations for future actions by the employers, social and health care providers, and researchers.  

The abstract offers an overview of the background, objectives, methods, results, conclusions, and references to the availability of concrete recommendations.

Introduction: offers an overview of the general context associated with the varied forms of sexual harassment, and of the scholarship examining the issue. The implications and the relevance of the study are well described.  

40-41: a suggestion to add references to the media and specify, whether this statement refers to Germany.   

166: it is hardly possible to obtain a complete picture of any relationship. Perhaps use “fuller/more comprehensive/more detailed/more nuanced” instead.

  1. Materials and Methods well describe the methodology and the sample. It also sets the grounds for raising the matter of a low response rate in the subsequent parts.
  2. Results offer a comparative overview of the outcomes, linking the level of awareness and the choice of resorting to the various forms of support offered.
  3. Discussion shows that the overall figures for healthcare and social services require separate studies in order to unmask significant sector-specific differences. Same applies to gender differences in the type of harassment experienced by the service providers.

Overall, the study represents an informative, well referenced read, that builds upon the already available research, while adding new insights. 

Reviewer 3 Report

This is a good paper, well written and structured. The methodology is appropriate, although it would be better to secure a more representative sample for such an important topic.  To my understanding, the paper is not publishable in its current form largely because of its Introduction.

  1. The authors write that the issue of sexual harassment of social and healthcare workers has been explored but at the same time they mention gaps. It is not clear what the gaps are, why they are so important to fill in, and why further exploration of the issue is timely and needed. I know that the authors tried to show that the study is timely but the way the Introduction is written has contradictions and, as a result, justification is not convincing.
  2. Is there a particular reason why it is important to explore this issue in Germany? Of course, social studies are usually geographically specific, but there should a better rationale for deciding to do this study in Germany. Also, how learning about Germany could help policy making and intervention elsewhere?

In summary, the paper lacks good justification.

Reviewer 4 Report

  1. The original sexual harassment scale has 14 items including nonverbal subscale (4 items), verbal subscale (6 items), and physical sexual harassment subscale (4 items). The reliability and validity of the original scale was not reported. In the current study, the authors arbitrarily reduced the verbal sexual harassment subscale from the original 6 items to 4 items. This is not recommended since validity depends on full and faithful use of the original format. Moreover, there is no rationale which 2 items should be deleted.

  1. The reduced sexual harassment scale was either used as a binary variable to calculate prevalence (If all items in a sexual harassment subscale were answered "never", the participant was considered to have not experienced this form of sexual harassment in the past 12 months) or as a continuous variable (mean value, such as in Figure 1). Which is the right way to use this scale? Moreover, the two ways did not yield consistent results in Table 3 and Figure 1. For example, the prevalence of verbal sexual harassment in hospital was the highest (75.9%, Table 3). In contrast, the mean value of verbal sexual harassment in hospital was lower than those in outpatient care service and inpatient care facility. It is very confusing.

  1. The authors concluded that “all three forms of sexual harassment showed substantial significant positive correlations, at relatively similar levels, with the impaired well-being of the individuals affected”. However, all the correlation coefficients were lower than 0.3 (ranged from 0.13 to 0.28), indicating weak correlations. Even though the correlation coefficients were significantly different from zero, all three forms of sexual harassment were weakly correlated with the impaired well-being.

  1. The authors stated that hierarchical multiple linear regressions were conducted and control variables included sex, age, contact with patients, clients, or residents, working hours, qualification, and leadership function. However, no results of the hierarchical multiple linear regressions were reported.

Round 2

Reviewer 3 Report

Thank you for considering my suggestions. The paper is in better shape now.